# Satellite Image Processing by Python and R Using Landsat 9 OLI/TIRS and SRTM DEM Data on Côte d’Ivoire, West Africa

**DOI:** 10.3390/jimaging8120317

**Published:** 2022-11-24

**Authors:** Polina Lemenkova, Olivier Debeir

**Affiliations:** Laboratory of Image Synthesis and Analysis (LISA), École Polytechnique de Bruxelles (EPB, Brussels Faculty of Engineering), Université Libre de Bruxelles, Building L, Campus de Solbosch, ULB–LISA CP165/57, Avenue Franklin D. Roosevelt 50, B-1050 Brussels, Belgium

**Keywords:** image processing, remote sensing, programming, Python, R, satellite image, digital terrain model, cartography, mapping, spatial analysis, 91.10.Da, 91.10.Jf, 91.10.Sp, 91.10.Xa, 96.25.Vt, 91.10.Fc, 95.40.+s, 95.75.Qr, 95.75.Rs, 42.68.Wt, 86A30, 86-08, 86A99, 86A04, Y91, Q20, Q24, Q23, Q3, Q01, R11, O44, O13, Q5, Q51, Q55, N57, C6, C61

## Abstract

In this paper, we propose an advanced scripting approach using Python and R for satellite image processing and modelling terrain in Côte d’Ivoire, West Africa. Data include Landsat 9 OLI/TIRS C2 L1 and the SRTM digital elevation model (DEM). The EarthPy library of Python and ‘raster’ and ‘terra’ packages of R are used as tools for data processing. The methodology includes computing vegetation indices to derive information on vegetation coverage and terrain modelling. Four vegetation indices were computed and visualised using R: the Normalized Difference Vegetation Index (NDVI), Enhanced Vegetation Index 2 (EVI2), Soil-Adjusted Vegetation Index (SAVI) and Atmospherically Resistant Vegetation Index 2 (ARVI2). The SAVI index is demonstrated to be more suitable and better adjusted to the vegetation analysis, which is beneficial for agricultural monitoring in Côte d’Ivoire. The terrain analysis is performed using Python and includes slope, aspect, hillshade and relief modelling with changed parameters for the sun azimuth and angle. The vegetation pattern in Côte d’Ivoire is heterogeneous, which reflects the complexity of the terrain structure. Therefore, the terrain and vegetation data modelling is aimed at the analysis of the relationship between the regional topography and environmental setting in the study area. The upscaled mapping is performed as regional environmental analysis of the Yamoussoukro surroundings and local topographic modelling of the Kossou Lake. The algorithms of the data processing include image resampling, band composition, statistical analysis and map algebra used for calculation of the vegetation indices in Côte d’Ivoire. This study demonstrates the effective application of the advanced programming algorithms in Python and R for satellite image processing.

## 1. Introduction

### 1.1. Background

Remote sensing (RS) data have great potential for geosciences as an important source of spatial information. The RS data can be used in a variety of applications, including geologic studies, land cover mapping, climate change detection and environmental monitoring. Semantic information obtained from satellite images is widely used in thematic mapping for analysis of the diverse phenomena and processes on Earth. Examples include mapping land cover fractions [1] and vegetation types [2,3,4,5,6], hydrologic and geomorphic modelling [7], urban sprawl and geological exploration [8,9]. Aside from satellite images, another source of RS data is provided by airborne images [10,11] and light detection and ranging (LiDAR) data for three-dimensional (3D) terrain analysis [12,13,14,15,16,17,18].

The general idea in image processing is that geospatial information can be derived from the interpreted RS data. For instance, repeated structures representing similar objects on the Earth’s surface are commonly depicted on satellite images, represented by pixels comprising the matrix of the image scene with a comparable range of values. These data can be visualised using the properties of the digital numbers (DN) of the pixels, which depend on the texture, composition and fabric of the objects shaping the face of the Earth [19]. Different land cover types which comprise the mosaic of the diverse landscapes can be detected on the images. Identifying and recognising these features is possible using the contrasting parameters of pixels in the satellite images: brightness, resolution [20,21,22], intensity [23,24] and texture [25,26,27]. Accurate and robust techniques of satellite image processing are essential for effective image interpretation in many approaches developed recently. These include diverse algorithms of image classification and processing.

The existing algorithms of image classification aim at reading the information from pixels through reading the values of raster cells. The analysis of this information enables mapping diverse phenomena, processes and objects based on the interpretation of RS data [28,29,30,31,32]. The main objective of image processing is obtaining this information from the RS data. In this context, the search for effective methods of image processing resulted in the variety of existing Geographic Information System (GIS) software designed for RS data processing. However, in many practical situations, the GIS presents limitations, either due to the commercial licencing of software with an associated high cost (ArcGIS, Erdas Imagine and eCognition) or functional restrictions (QGIS and the System for Automated Geoscientific Analyses (SAGA) GIS). Moreover, the traditional manual methods of image processing are time-consuming, which make them ineffective for big data processing and require automation.

### 1.2. Related Work

Programming languages present effective solutions for processing satellite images. This is achieved through the advanced methods of machine learning (ML) [33,34,35,36]. Among them, scripting techniques enable handling RS data effectively and accurately [37,38,39,40,41,42]. An essential advantage of the programming languages consists of automation of the RS data processing. A second advantage consists of the compatibility of data formats. In contrast to the stand-alone GIS software, scripting libraries can be integrated with other GISs via modules, plugins and toolkits [43,44,45]. A variety of existing programming languages can be used as optimal and effective tools for image processing and RS data analysis in environmental modelling, among which Python and R are the most advanced and widely used.

Python is undoubtedly one of the most successful high-level programming languages [46]. It has become one of the most powerful tools that can be used for data science. Python enables performing various tasks of image processing and spatial analysis through integration of libraries [47,48,49]. Recent examples include image interpolation, calibration and correlation [50], compressed sensing and image reconstruction [51], identifying and extracting vertical features [52] and meteorological modelling [53] with resulting images in compatible formats.

A collection of Python libraries [54,55] has recently built up with EarthPy that is specially tailored for geo-information processing [56]. Aside from general applications in diverse branches of data science and ML [57,58,59,60], Python can be applied for RS data processing using its spatial libraries [61]. Image and signal processing by its object-oriented algorithms of Python are pervasive in geosciences and provide solutions to a wide range of problems. Examples of using Python have been reported in a variety of domains, including geodetic studies [62], topographic analysis [63], hydrogeological modelling [64], modelling air pollution and land use [65], batch spatial data processing [66], creating panorama images [67] and tunneling microscopy [68], to mention a few.

R is a fundamental programming tool for statistical and spatial data processing [69]. It includes a variety of packages with extended graphical functionality and advanced statistical components [70]. The packages of R have long been used for data retrieval, analysis, modelling and visualisation in various industries including robotics, engineering and GIS [71,72,73,74,75]. R demonstrates superiority in statistical computing, graphical plotting and data analysis compared with graphical user interface (GUI) software. Moreover, R excels in big data analysis [76,77,78], data mining [79] and visualisation [80] modelling, plotting and image processing, which are supported through the variety of its built-in packages.

Compared with Python, R has a richer collection of libraries due to the long history of stable development. They contain a wide variety of mathematical statistical tools, from the very basic functions to the highly sophisticated algorithms of data analysis [81]. Recent advances in geospatial data processing [82] and image classification [83,84] have made R successful for remote sensing data processing, including image classification and segmentation. This poses a new way to process and model satellite images and derive spatially structured information from complex scenes. In this regard, the ’terra’ and ’raster’ libraries of R present essential tools for modelling geospatial data. The functionality of these libraries includes essential options for image classification, such as reading spatial dimensions from raster images, supporting spectral bands of Landsat scenes, evaluating the similarity between the values of pixels and extracting features from variables such as vegetation types and variations in landscape structure and topography.

### 1.3. Contribution

The goal of this paper is to present the application of Python and R algorithms for image processing. This work is motivated by the idea of using advanced programming methods to visualise object features and classes on the Earth’s surface. The specific objectives include the following tasks: (1) plotting the colour composite of a Landsat Operational Land Imager and Thermal Infrared (OLI/TIRS) image by combinations of the three channel sensor bands to display the image; (2) computing the selected vegetation indices using the principles of map algebra based on the DN of reflectance value of the pixels in the selected spectral bands of near infrared (NIR) and red; and (3) terrain modelling with geomorphic visualisation of the slope, aspect, hillshade and elevation for relief analysis using the data derived from the Shuttle Radar Topography Mission (SRTM) digital elevation model (DEM).

We propose two approaches to satellite image processing by Python and R. In the first approach, we use R packages ‘terra’ and ‘raster’ for calculation of the vegetation indices of Landsat bands and creating colour band composites. For the second approach, we present an application of the Python library EarthPy for terrain analysis supported by auxiliary libraries (Matplotlib, NumPy, Rasterio and Plotly) for data processing and visualisation. The combination of several libraries has been demonstrated to be effective in utilising the properties and functionality of each to achieve effective image processing for geospatial analysis using the Python syntax.

The objectives of our study are to use the advanced functionality of R and Python for satellite image analysis aimed at the environmental assessment and mapping of Côte d’Ivoire using broadband multispectral images. Here, the goals are to identify the vegetation conditions by computing four vegetation indices—NDVI, SAVI, EVI2 and ARVI2—and to model the terrain in the surroundings of Kossou Lake to the north of Yamoussoukro. Several libraries are used in scripts for terrain modelling, computing the vegetation indices and creating colour composites using a combination of Landsat bands. In contrast to the existing approaches, which use the traditional GIS software with restricted functionality, our method applies flexible algorithms of scripting languages using the syntax of Python and R and extended functionality of their libraries.

We find that the advanced libraries of Python and R are effective for processing high-resolution satellite images. This motivates us to introduce an efficient scripting method to environmental monitoring. To the best of our knowledge, no attempts have been made to integrate high-level languages, such as Python and R, for mapping Côte d’Ivoire, for terrain modelling or for vegetation analysis by scripts. Works related to ours include some previous studies on Côte d’Ivoire [85,86,87,88,89], in which traditional GIS methods were applied to process spatiotemporal information for environmental mapping. The key difference between our work and theirs is that our method integrates programming algorithms for processing multispectral satellite images using scripts.

Our main contributions can be summarised as follows: (1) environmental mapping of Côte d’Ivoire based on modelling and visualisation of several vegetation indices using the ‘terra’ package; (2) an R-based approach to processing satellite images for automatic processing of raster data in spectral bands and computing cell values aimed at spatial analysis of the heterogeneous landscapes of Côte d’Ivoire; (3) EarthPy-based algorithms for deriving topographic information of the Kossou Lake region to model the terrain parameters of the slope, aspect, hillslopes and stream channels of the hydrological network system; and (4) mapping spatial data with scripting libraries for automatic data processing, which outperforms the traditional approaches for cartographic visualisation and image analysis.

## 2. Study Area

The study area was focused on the central part of Côte d’Ivoire, West Africa (Figure 1). In particular, we selected the region of Kossou Lake as an enlarged fragment of the study area for terrain analysis. This lake was formed artificially by a constructed dam on the Bandama River. Kossou Lake forms one of the most important inland water bodies of Côte d’Ivoire and plays an essential role in the environment of Sub-Saharan Africa [90]. It has functional aspects as a habitat for dominant biotas and supports food chains in ecosystems.

Côte d’Ivoire is notable for a diverse terrain structure which has two distinct parts: a coastal region formed by the alluvium and marine sands and a continental region with hilly relief [91]. The land cover types are associated with two major ecosystem types: forests distributed in the southern part of the country and savannah to the north off Bouaké [92]. The vegetation types are largely controlled by the equatorial or subequatorial rainfall regime. A long dry season is typical for the northern half of Côte d’Ivoire with dominating savannahs, while a tropical regime characterises the southern part of the country with prevailing forests [93]. A favourable climatic and environmental setting maintains the farming and agricultural activities of the local population. This makes Côte d’Ivoire the largest cocoa producer in the world [94,95,96,97] and an active exporter of food-producing crops [98], such as coffee, bananas, pineapples and palm oil [99]. Aside from agriculture, the activities of population include exploration, mining [100,101,102] and limited fishing along the narrow continental shelf of the Atlantic Ocean [103].

Recently, Côte d’Ivoire experienced environmental changes typical for other countries of humid West Africa. These include deforestation and fragmentation of ecosystems [104], a decrease in species richness in the savannah biomes [105] and an increase in habitat heterogeneity, which affects biodiversity and the species community structure [106]. The triggers for the environmental changes include climate effects and urban sprawl, which has been exceptional for Côte d’Ivoire since the 1950s, due to the natural demographic growth and the increase in industrial agglomerations [107,108,109,110]. Geographically, the acceleration of the population leads to increased city areas and modified transport networks [111]. Other environmental problems include water supply issues and pollution caused by agro-industrial plantations [112].

## 3. Materials and Methods

### 3.1. Data

The data were collected from the United States Geological Survey (USGS)’s Global Visualization Viewer (GloVis), a satellite image repository with publicly available open access to remotely sensed data (Figure 2).

The data used in this study included the satellite imaging of Landsat 9 OLI/TIRS C2 L1 and the SRTM DEM for terrain modelling. The satellite image covers the central part of Côte d’Ivoire. It has the following technical parameters: a LC91970552022011LGN01 scene identified; Landsat product ID of LC09_L1TP_197055_20220111_20220122_02_T1 and land cloud coverage of 1.29, acquired on 11 January 2022. The image was obtained from the Landsat 9 Earth observation satellite, which has two remote sensing instruments for optical and thermal sensors onboard: the Operational Land Imager (OLI-2) sensor and the Thermal Infrared Sensor (TIRS-2) along with nine spectral bands [113,114]. More technical information is available online from Landsat 9. The data used for terrain analysis included the SRTM void-filled DEM, obtained from the SRTM3 NASA/NGA. It has a resolution of 3 arc-seconds (90 m) and the following technical parameters: an entity ID of SRTM3N07W006V2 and tile coordinates: of 5–6° W, 7–8° N, with an acquisition date of 11 February 2000. The general map showing the study area was plotted based on the General Bathymetric Chart of the Oceans (GEBCO) [115] using GMT scripting [116] and following the existing methodology [117,118].

### 3.2. Image Processing in R

The R packages ‘terra’ [119] and ‘raster’ [120] were used for computing the vegetation indices and image processing. The colour scheme resource of the RColorBrewer package was used for visualisation and plotting [121].

#### 3.2.1. Information Retrieval and Spatial Statistics

To cope with a series of Landsat bands and search for metadata, we examined the image parameters using regular expression with the command line utilities of Unix as a combination of grep, the Geospatial Data Abstraction Library (GDAL) and echo (Listing 1). Here, grep searches for text in a file according to a given pattern of characters, the GDAL filters spatial information in the dataset, and echo outputs the strings using the defined arguments.

**Listing 1:** Unix shell script by GDAL and echo-grep utilities to inspect the geometry of Landsat TIFs.

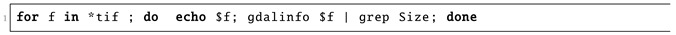



The spatial resolution, corresponding to the pixel size, was 30 m for Bands 1–7, 9 and 10, while it was 15 m for Band 8 (panchromatic) (Figure 3a). The spatial information of the Landsat channels (bands) is presented in metadata, as shown in Figure 3b, and inspected by R. It indicates the number and location of the spectral measurements, spatial extent, geodetic information (coordinate reference system (CRS), ellipsoid and datum obtained by the global positioning system (GPS) of the spacecraft and the spatial and radiometric resolutions of the image. Each image corresponding to a given band of the Landsat scene was read into the R SpatRaster object class, which represents the equalised area of the rectangular cells with regard to the units of the CRS (Figure 3b).

#### 3.2.2. Spectral Bands for Colour Composites

The properties of the image were evaluated using a SpatRaster object by the R commands presented in Listing 2. Here, we inspected the CRS, the geometry of the image (number of cells, rows and columns) and extent, resolution and origin of the bands. The panchromatic channel (Band 8) had a higher resolution (15 m) compared with the set of the multispectral bands in the visible and thermal bands of the spectra (30 m). To ignore this difference in the matrix of the image, the raster in Band 8 (panchromatic) was resampled to the structure of the same number of rows and columns as in other bands (Here, target Band 1 is in Ultra Blue) by creating the RasterBrick multi-layer object as demonstarted in Listing 2.

**Listing 2:** R code used for inspecting the information and statistics on the Landsat 8-9 OLI bands.
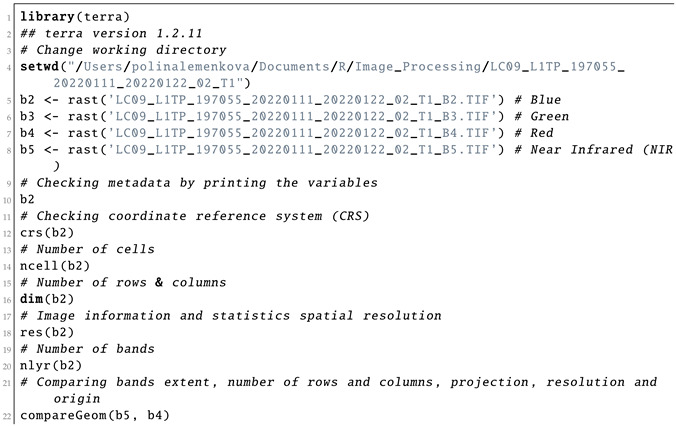

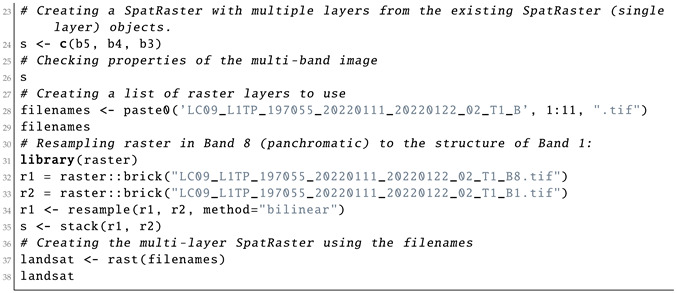


Afterwards, the layers were read in the ‘landsat’ object class of R, where they were represented by the bands of the Landsat-9 OLI image. The reflection intensity had the following wavelengths: Ultra Blue or coastal aerosol (Band 1), blue (Band 2), green (Band 3), red (Band 4), near infrared (NIR) (Band 5), Shortwave Infrared (SWIR) 1 (Band 6), Shortwave Infrared (SWIR) 2 (Band 7), panchromatic (Band 8), cirrus (Band 9), Thermal Infrared (TIRS) 1 (Band 10) and Thermal Infrared (TIRS) 2 (Band 11). The visualisation of single bands was performed using Listing 3 and is presented as a subplot of grayscale scenes in Figure 4.

**Listing 3:** R code for plotting the 11 original individual layers (raw bands) of the Landsat-9 multi-spectral images as grayscale bitmap images.

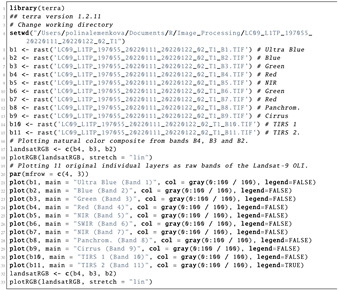



There was a difference in nuance between the grey corresponds and the numerical values and range of pixel values in diverse channels of Landsat. This is explained by the spectral reflectance that differs in various bands for objects on the Earth’s surface. Due to the individual reflectance of solar radiation by these features, the values differed accordingly. Such effects from spectral reflectance enabled us to use combinations of various bands to better represent the diverse phenomena and objects on Earth. For instance, the particular applications of Landsat bands are for geology (SWIR (Band 7), NIR (Band 5) and red (Band 4)) [122], mineralogical identification of soil and rock (SWIR (Band 7), red (Band 4) and green (Band 2)) and identifying the difference between soil and vegetation (green in Band 3) [123,124]. Aside from that, the reflectance data from the selected bands of the Landsat images were used in the calculation of four vegetation indices for the study area using the interactions of pixels in the raster.

Subsetting large files, such as multispectral Landsat images, was performed using the SpatRaster object and ’subset’ function, which enabled processing the necessary bands of the Landsat image. In this respect, it enabled ignoring the cirrus, TIRS and other bands that were not relevant for a given task and to leave only the visible spectra. In this way, a subset of the target layers was selected from the SpatRaster object, while other bands were removed from the dataset using Listing 4.

**Listing 4:** R code for subsetting the large files using SpatRaster object.

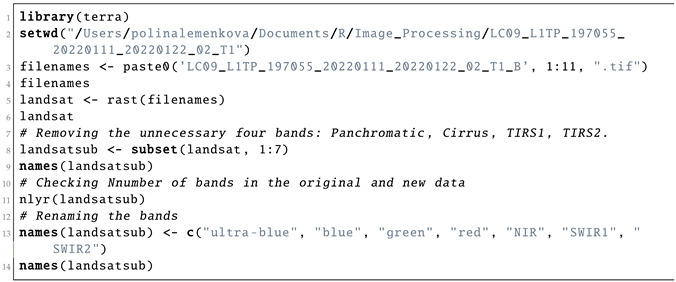



Visual inspection of the satellite image product built from the monochrome image data (Figure 4) was useful for extracting general information on the large-scale features, such as river systems, lakes, mountains, fracture patterns, large geological lineaments or the extent of the city with notable objects such as highways. These objects are visible and can be identified by their shape and size and interpreted using spatial properties related to the pixel brightness values.

To more thoroughly investigate the relationship between the spectral responses of various land cover types in different wavelength diapasons, we examined the pairwise correlation between the selected bands of the Landsat image. To this end, we compared the bands in pairwise combinations (Band 1 against Band 2 and Band 4 against Band 5) in the selected parts of the spectrum. The images were plotted pixel by pixel against those from the corresponding bands using Listing 5. The resulting plot showed a narrow correlation ellipse and a histogram representing a strong positive correlation between the two bands of green and blue as well as red to NIR (Figure 5).

The cells with low spectral reflectance in a target band generally had low reflectance in the remaining bands, which was caused by the albedo of the Earth’s surface. However, a visible correlation between Bands 1 and 2 showed more similarity between these bands, as all the pixels were located almost along the same line (Figure 5a). In contrast, the NIR and red bands demonstrated more differences in spectral reflectance for the selected objects, which resulted in a scattered cloud of points representing the pixels of the image (Figure 5b). The brightness of the pixels in each layer indicates the amount of incident solar radiation, which is reflected by the Earth’s surface in a given wavelength diapason. As a result, this causes a similarity or difference in the spectral bands of the Landsat scene.

**Listing 5:** R code for inspecting the correlation between the selected Landsat bands.

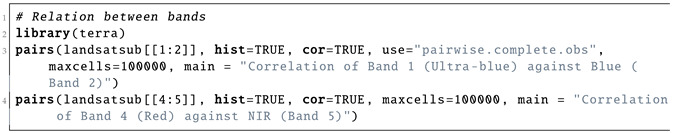



Since there was not much information in the monochrome bands of the Landsat OLI/TIRS image, they were combined into colour composites to create more representative images. Plotting of the colour composites was based on the combinations of the three bands used as elements of the matrix scene. The three bands assigned to each of the three primary colours (red, green and blue (RGB)) were displayed with images using Listing 6.

**Listing 6:** R code used for plotting the color composites for bands of the satellite image.
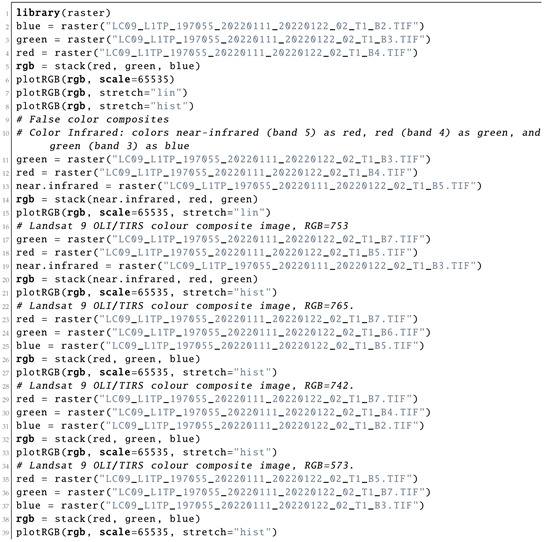

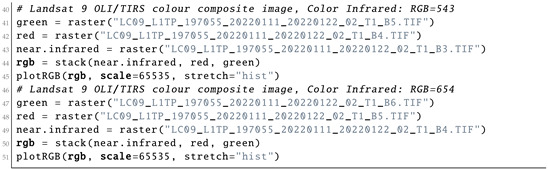


The vegetation indices were computed based on the arithmetically transformed combination of Landsat OLI/TIRS bands. This resulted from the existing formulae for band computations using mathematical operations with bands. The map algebra included addition, subtraction and division of the brightness in cells of the NIR and red bands of the Landsat image. The differences in the bands highlight the regions of the enhanced and more healthy vegetation in the target area of the image, which presents information for environmental monitoring. The resultant image shows brightness values of vegetation modified by colour palettes made using arithmetic operations on the composite bands.

The most widely used and well-known vegetation index is the Normalised Difference Vegetation Index (NDVI), which applies the ratio of the difference in values between the red and NIR bands and their sum. Similar to the NDVI, the Enhanced Vegetation Index (EVI) quantifies vegetation greenness with added corrections for the atmospheric conditions and canopy background noise. Aside from that, the EVI is more sensitive in forest areas with more dense canopies and vegetation coverage, which is useful for the forest regions of Côte d’Ivoire. The Soil-Adjusted Vegetation Index (SAVI) is built up by using the NDVI and correcting for the effects of soil brightness in the regions where vegetative cover is low, namely deforested areas and the fragmented savannahs of Côte d’Ivoire. The Atmospherically Resistant Vegetation Index 2 (ARVI2) is the most applicable index in regions with high atmospheric aerosol contents. Examples of other environmental indices include the Normalized Difference Water Index (NDWI), modified Normalized Difference Water Index (MNDWI) and Water Ratio Index (WRI) [125].

Computing the vegetation indices is an important environmental tool which is based on information from the DN values of pixels, indicating the spectral reflectance of the red and NIR bands as measured by the sensor. It is commonly computed with the spatial statistics and algebra of the bands using the cell values of the image as shown in Listing 7. The reason for selecting the NIR and red bands for computing the vegetation indices is vegetation appearing brighter in these bands, since it reflects more energy in the NIR and red bands compared with other wavelengths. Contrarily, water absorbs most of the energy in the NIR band and is therefore represented by black to very dark brown colours. Through a series of interactions among the elements of an image, they are allocated to the classes based on the weighted values of their relative proportions.

**Listing 7:** R code used for computing vegetation indices from bands of the satellite image.
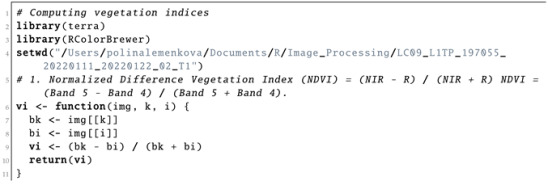

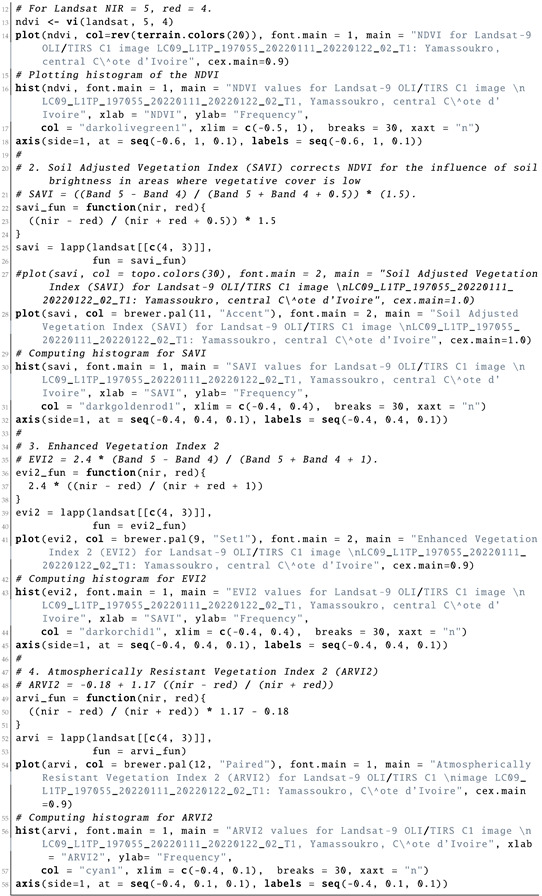


### 3.3. Terrain Analysis in Python

To model the relationship between the topographic variables and appearance, we applied Python algorithms for processing SRTM data by the 3D approach. Two different types of local features were computed for the SRTM image, namely the DTM and hillshade, with artificial light imitating the Sun’s illumination. The 3D visualisation of the terrain surface is changed by varied azimuth and altitude of sun. Technically, the terrain analysis has been performed using integration of several libraries of Python: EarthPy [56] for modelling, Matplotlib for data visualisation [126], Rasterio for processing gridded raster datasets [127]. The auxiliary packages include NumPy [128] which operates with arrays of raster cells presenting the elements of the matrix of image, arranged in regular rows and columns, and Plotly [129], which is used as an additional graphical and plotting library.

The geospatial TIFF was visualised for modelling the DEM as GeoTIFF using the Rasterio library, which imported the dataset as a raster array of cells, read the GeoTIFF format and stored a gridded raster as a digital terrain model (DTM) or satellite images (Listing 8). The elevation data were imported by Rasterio as height values that were contained in the cells of the corresponding raster. The structure of the data presented a NumPy array in the form of a matrix with the DNs of pixels. The terrain was plotted as an array of cells representing the land surface according to the elevation values, as shown in Listing 8. Plotting of the bands was performed using the ep.plot_bands() function using the spatial dimensions of the raster file.

**Listing 8:** Python code (1) used for plotting DEM in Figure 2.

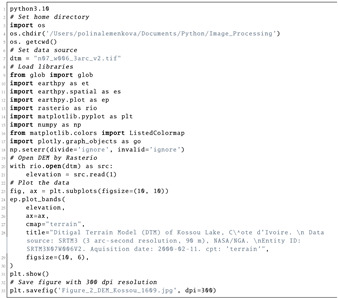



The hillshade was calculated using the altitude of the Sun as a light source at the zenith and azimuth of the illumination source (Listing 9). The functions to compute the variations between the exemplar locations of artificial illumination of the terrain resulted in the visualised details of the terrain in the Kossou Lake surroundings. The slope and aspect of the terrain were used as auxiliary variables for the terrain modelling. The hillshade modelling was inherently related to static cartographic representations of the Earth’s topography, and thus hillshade was used for the geomorphometric representation of the terrain of Côte d’Ivoire.

**Listing 9:** Python code (2) used for plotting hillshade in Figure 3.

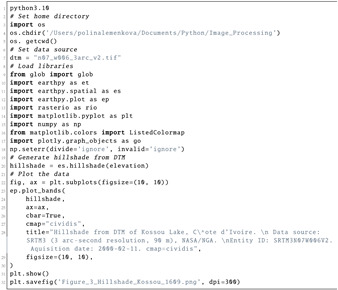



Hillshade represents the terrain with a hypothetical illumination value in the monochrome scale (0–255) with cells on the raster scene showing the terrain surface with a given specified light source. The illumination was changed to 10° and 230°(in radians) to better represent a geospatial model of the topographic relief of Côte d’Ivoire (Listing 10). Assume that the orientation differences between the angle and azimuth of artificial illumination in modelling the topography of Kossou Lake are independent and identically distributed according to the time and seasonal parameters. To visualise the characteristics of the relief in the topographic model based on the DEM, raster data were sampled to estimate the variations in distribution of the parameters. In our case, we selected azimuths of 10° and 230°. We observed that the effects from the contrasting variations in relief approximately followed the changes in the Sun’s azimuth and angle altitude.

**Listing 10:** Python code used for plotting the hillshade with changed azimuth and Sun angle values for Figure 4 and Figure 5.
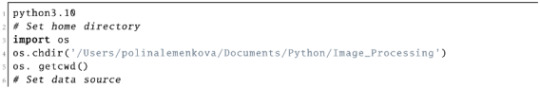

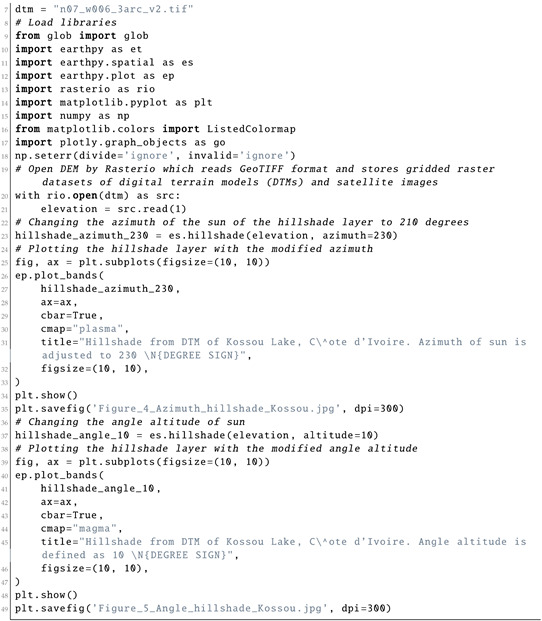


We assumed the data to be of a high resolution, which may have presented our current limitations. However, the precision may affect the resulting topographic model through coarser resolution data (i.e., lesser meters in one pixel will give a rough representation of the Earth’s surface and miss smaller landforms in the landscape). Regarding this, it is strongly recommended to apply very high-resolution data for a precise representation of the Earth. Another point of concern consists of selecting the correct view shed and azimuth for modelling the aspect, slope and elevation, which is especially true for mountainous regions and rugged reliefs. Although the Python-based method of terrain modelling can potentially handle materials with coarser or unknown resolutions for the original data, the results may hide the selected structures on the Earth’s surface. The same issue concerns the applications of R for image processing, where the precision of the output data corresponds to to the original resolution of the input data. In our future works, we are very interested in extending our study to handle higher-resolution imagery, such as Sentinel 2A at a 10 m resolution or Satellite pour l’Observation de la Terre (SPOT) with up to a 1.5 m spatial resolution in the panchromatic and multispectral channels.

In the presented methodology, we applied Python and R algorithms which process and model the terrain characteristics of the Earth using the spatial parameters of the remote sensing data. By exploiting the spectral reflectance characteristics of the satellite bands, we used information from the Landsat channels from the corresponding cells which represent objects located on the Earth. Interpretation of these values is based on their spectral reflectance in various wavebands, which provides information on the characteristics of typical land cover types. This was possible through the analysis of the different brightness of each land cover type (e.g., forests, rivers, lakes and other water bodies or urban spaces). In turn, the geomorphometric models formed using Python scripts enabled predicting the distribution of various soil properties and associated vegetation types which were related to the specific slope steepness and regional topographic features. Examples of the application of terrain models include hazard risk assessment in applied geology for erosion, landslides and avalanches in mountainous regions. The major advantage of using Python and R scripts for environmental modelling is that it reduces the time of data processing and increases the precision of models through automation of data processing, because scripts are repeatable for other regions and can be applied in similar studies.

## 4. Results and Discussion

### 4.1. Color Composites

Colour image maps were created by different combinations of the recorded Landsat OLI/TIRS bands. In the sequence from top to bottom, they are displayed as natural and false composites, respectively (Figure 6). The use of natural (or true) colour composites was widely applied for general environmental monitoring, since it exposed characteristics regarding the landscape and land cover types compared to the monochrome images. Thus, band combinations 4-3-2 (red-green-blue) and 2-3-4 (blue-green-red) represented natural colour composites well. Since its appearance is similar to the normal aerial photograph, this composite is advantageous for mapping vegetation and agricultural coverage and discriminating the vegetation from bare soil and growing crops (Figure 6a,b).

In Figure 6c, Bands 3,4 and 5 were taken in the inverse way, where Band 5 (NIR) represented red, Band 4 (red) was green and Band 3 (green) was blue. The improvement of the image was performed using linear and histogram stretching, which consisted of the adjustment of the contrast in the colour tones of the images. In this band composite, the broadleaf healthy vegetation had a deeper reddish hue, while lighter hues were related to the grasslands, savannahs or sparsely vegetated areas. Soil was represented by various hues, depending on the soil type, from dark to light browns. The populated urban areas were shown in light grey to white spots.

The false colour composite was represented by the mix of bands 3, 4 and 5, where the NIR, red, and green bands were used. This printed the optical data, with vegetation dominating through the bright red hues due to the chlorophyll reflection, while agricultural lands were depicted in cyan. The bare soils in this combination appeared as shades of white to light grey and blueish or green in the agricultural regions, depending on the crop type. The darker shades of each colour generally indicated moister soil and cities with hues of steel grey (Figure 6c). The combination of Bands 7,6 and 5 showed the mix of SWIR 2, SWIR 1 and NIR. Here, the crop lands were coloured yellow, while vegetation areas were shown as bright ultramarine blue colour (Figure 6d).

Figure 7 shows different cases of false colour composites mixed using NIR, SWIR2 and green (a), SWIR 2, red and blue (b), NIR, red and green (c) and SWIR 1, red and green (d). Since NIR light is reflected from the surfaces with higher levels of vegetation but absorbed by water, such band combinations are useful for vegetation and agriculture applications (e.g., mapping forests and crops).

### 4.2. Computing Vegetation Indices

#### 4.2.1. Normalized Difference Vegetation Index (NDVI)

The NDVI aims at quantifying the concentrations of the green leaf canopy. It indicates healthy, vigourous vegetation by using the arithmetic of the red and NIR bands as shown in Equation (Equation 1) [130]:(1)$$\mathrm{NDVI}=\frac{\left(\mathrm{NIR}-\mathrm{Red}\right)}{\left(\mathrm{NIR}+\mathrm{Red}\right)}$$

Originally tested in 1974 [130], the NDVI received wide recognition in image processing for environmental modelling. Nowadays, the NDVI is the most famous and most used vegetation index. The selection of the red and NIR bands in the NDVI is explained by the specific certain abilities of plants to selectively absorb, reflect or transmit the frequencies in the wavebands; plants absorb red and strongly reflect NIR waves. Practically, the NDVI enables indicating the green vegetation density covering the terrain surface and identifying where plants are thriving and where they are under stress (i.e., due to a lack of water).

Due to this fundamental property, the NDVI was used to evaluate vegetation health and assess the vigour of plants in agricultural regions. The general NDVI lies in the diapason from −1 to +1, but the majority of the values are restricted by a range from −0.1 to 0.5 (Figure 8). Negative values for the NDVI correspond to water or very moist soil areas, as shown by the beige-to-white colours in Figure 8, while higher NDVI values signify a dense vegetation canopy, which is depicted by the bright green colour in Figure 8.

#### 4.2.2. Soil-Adjusted Vegetation Index (SAVI)

The Soil-Adjusted Vegetation Index (SAVI) is a distance-based vegetation index which is based on the NDVI but includes an adjustment factor to reduce the noise from the NDVI by eliminating the effects from the canopy background and atmospheric conditions. Thus, it adjusts the NDVI for soil brightness where vegetation coverage is low in urban, densely built up areas and bare soil using the arithmetic expression in Equation (Equation 2), proposed in [131]:(2)$$\mathrm{SAVI}=\frac{\left(\mathrm{NIR}-\mathrm{Red}\right)}{\left(\mathrm{NIR}+\mathrm{Red}+L\right)}\times \left(1+L\right)$$
where L is a soil brightness correction factor, which is defined as 0.5 to adjust to most land cover types. Compared with the NDVI, the SAVI is a finer indicator, which enables it to be used for the assessment of if land coverage contains healthy green vegetation with correction for brightness, with the majority of values lying in the diapason from −0.1 to 0.1 (Figure 9). Thus, the main advantage of the SAVI is the emphasised colour of the vegetation canopy corrected for the soil brightness, because the SAVI excludes the atmospheric or meteorological effects and those from soil.

Compared with the NDVI and EVI, which were affected by the soil brightness, the SAVI demonstrated better results in differentiating between vegetation and soil. Thus, the SAVI index was more suitable and better adjusted to agricultural monitoring. At the same time, because Côte d’Ivoire is a highly productive agricultural region, being the world’s most important cocoa producer, the agricultural monitoring of its landscapes is of high importance. Therefore, analysis of the vegetation indices is valuable for sustainable development and environmental planning in Côte d’Ivoire.

#### 4.2.3. Enhanced Vegetation Index 2 (EVI2)

The EVI (Figure 10) is an optimised spectral imaging transformation of the NIR and red image bands aimed at enhancing the contribution of vegetation properties. Computing this index is performed using Equation (Equation 4), originally proposed by [132].

EVI2 was developed as an updated and adjusted formula of the EVI. In this form, it was developed as a satellite vegetation index for the Moderate Resolution Imaging Spectroradiometers (MODIS) and adopted for the Landsat Thematic Mapper (TM) or Landsat Enhanced Thematic Mapper Plus (ETM+) series. The expression is defined as follows in Equation (Equation 3), which was optimised in [132]:(3)$$\mathrm{EVI}2=2.4\times \frac{\left(\mathrm{NIR}-\mathrm{Red}\right)}{\left(\mathrm{NIR}+\mathrm{Red}+1\right)}$$

This allows a robust spatial and temporal correlation of the terrestrial chlorophyll and photosynthetic activity, which affects the structure of the canopy in various types of vegetation, including savannahs and forests. The ratios of the NIR and red spectral bands in the image enable decreasing the topographic effects from the Earth’s relief. Therefore, the visualised EVI highlights and stresses smaller differences in the characteristics of spectral reflectance between vegetation and all other land cover types. The range of the values normally lies in the diapason between −0.2 and 0.2, as can be also seen in Figure 10:(4)$$\mathrm{EVI}=\frac{\left(\mathrm{NIR}-\mathrm{Red}\right)}{\left(\mathrm{NIR}+C1\times \mathrm{Red}-C2\times \mathrm{Blue}+L\right)}$$

#### 4.2.4. Atmospherically Resistant Vegetation Index 2 (ARVI2)

ARVI2 is adjusted to be resistant to atmospheric effects. At the same time, it is more sensitive to a wider spectrum of concentrations of chlorophyll in leaves. Specifically, for Côte d’Ivoire, the values of the ARVI ranged from −0.28 to −0.1 (Figure 11), although in general, the diapason of the ARVI values may vary from −1 to 1 by following Equation (Equation 5), which was originally developed in [133]:(5)$$\mathrm{ARVI}=\frac{\left(\mathrm{NIR}-\mathrm{Red}-y(\mathrm{Red}-\mathrm{Blue})\right)}{\left(\mathrm{NIR}+\mathrm{Red}-y(\mathrm{Red}-\mathrm{Blue})\right)}$$
where y is the quotient derived from the components of atmospheric reflectance in the blue and red channel. The equation for the adjusted ARVI (ARVI2), which is more sensitive to vegetation fraction, is presented in Equation (Equation 6), which was originally developed in [133]:(6)$$\mathrm{ARVI}2=\frac{\left(\mathrm{NIR}-\mathrm{Red}\right)}{\left(\mathrm{NIR}+\mathrm{Red}\right)}\times 1.17-0.18$$

This shows the regions of high atmospheric aerosol contents with higher moisture. Compared with the NDVI, the ARVI adjusts and minimises the atmospheric scattering effects due to the blue light reflectance values, which also affects the reflectance in the red light [133].

Similar to the NDVI, ARVI2 is also sensitive to the vegetation ratio in the landscape and useful in climate and environmental studies. Specifically, it can be applied to evaluate the absorption of solar radiation, which is highly applicable in tropical African regions such as Côte d’Ivoire.

### 4.3. Terrain Analysis

Cartographic visualisation is an essential application of satellite image analysis as a geospatial dataset. In this regard, terrain mapping was performed using remote sensing data and the integrated approach of Python and R (Figure 12).

The data were linked with the digital elevation model (DEM) to demonstrate the effective approach of both languages to modelling Earth observation data. We selected a DEM with a high spatial resolution for terrain modelling because it is one of the most crucial base layer factors for environmental modelling [134]. The advances in cartographic modelling using programming scripts in recent years changed the terrain visualisation from GUI-based GIS mapping to dynamic scripts with automatic representation of the output images. Processing the DEM from the SRTM using Python scripts based on the EarthPy library demonstrated the advantages and flexibility of the workflow, aimed at deriving topographic information for spatial analysis (Figure 13). The DEM developed from the SRTM of rugged terrain in Côte d’Ivoire had a spatial resolution of 3 arc seconds, which corresponds to ±90 m.

The topographic variation was modelled with the Python library EarthPy, based on the processing of fields (raster layers), spatial entities (elevated objects with 3D coordinates as attributes) and the network (topology of spatial links among the geographic objects). This approach uses relational data modelling with height as the primary identifier in a dataset. The DEM data are modelled by EarthPy using a raster data structure as a square grid matrix, with the topography defined by the sampled points and with XYZ dimensions indicating the continuous field of the spatial altitudes.

Spatial modelling continues topographic representation using discrete boundaries (Figure 13). It is based on the topographic attributes derived from the DEM and aims to define the geomorphic and hydrological features as environmental gradients. These derivatives include such attributes as the slope, aspect, upslope drainage and catchment areas, tangential, plan and profile curvature of the cross-sections, upslope and flow path length and hillshade. Determining the drainage direction and connectivity of the raster points from the DEM depends on the surface attributes of the topographic hillslopes and stream channels which comprise the hydrological network system. In this regard, morphometric modelling contributes to landscape analysis through visual representation of spatial variability and connectivity of the primary terrain attributes distributed on the surface of the Earth.

Numerous applications of topographic modelling in Earth and environmental studies motivated the development of Python programming approaches aimed at calculating the terrain. This is especially advantageous for the topographically diverse landscapes with interspersed forest and savannah areas which are typical for West Africa. The principal advantages of the EarthPy library consist of straightforward and robust topographic modelling, which presents the visualised series of the topographic attributes derived from a square grid DEM. These include the slope, aspect and topographic relief shading with their attributes (i.e., hillshade with varied Sun azimuths and angle altitudes), as demonstrated in Figure 13.

The environmental effects from the terrain landforms are reflected in the soil characteristics, as its formation is regulated by a variety of factors: the geologic substrate, direction of stream flow and susceptibility of hillslopes to soil erosion. In turn, this depends on the slope steepness and relief curvature. Application of the terrain model includes analysis of the sediment transport, which is controlled by the terrain type and contributes to soil erosion. That aside, the morphometric aspects, differences in topographic heights, disposition of the slope and aspect and topographic horizon affect the distribution of solar radiation. The latter controls the soil vegetation system through the biophysical processes of evapotranspiration and photosynthesis. Thus, the disposition of a hill slopes towards the sun angle, as the slope aspect contributes to the distribution of vegetation types through the effects of microclimate. For instance, more sensitive plants grow on the southern flanks of the hill slopes to benefit the solar energy. Thus, spatial continuity of the environmental processes is deeply linked to the regional topographic properties of the terrain. Therefore, modelling the topographic attributes computed by the Python library EarthPy, applied to a high-resolution DEM, reveals the variability of the terrain as a complex heterogeneous structure and contributes to the environmental analysis of Côte d’Ivoire, West Africa.

## 5. Conclusions

In this paper, we proposed two scripting approaches to satellite image processing to compute several vegetation indices and to perform terrain modelling. The scripting techniques of the ’raster’ and ’terra’ packages of R for computing vegetation indices demonstrated great potential for using map algebra and calculation techniques with the Landsat OLI/TIRS for monitoring vegetation health and environmental mapping. Moreover, this technique is also useful for agricultural applications where deteriorated sites may require remediation or close monitoring. To model terrain variables, we proposed a Python-based approach where the topographic data are derived from the SRTM DEM using the EarthPy library.

The problem of the terrain analysis in image processing can be referred to as spatial data modelling, with classification of cells according to the elevation values. The derived variables are essential for visualisation of the slope, aspect and hillshade using the parameters of the Sun’s angle and azimuth. Mapping the topographic data using terrain analysis by Python highlighted their fundamental importance in Earth modelling. Thus, the relief of the Earth’s surface has notable effects on various components of the ecosystems, including geological, climatic and hydrologic processes as well as the soil vegetation parameters. Moreover, topography regulates and controls climate processes such as precipitation, wind direction and strength. Therefore, it plays a crucial role in the environmental dynamics and ecosystem functioning, which includes the uneven distribution of the diverse landscapes with contrasting types of vegetation in the hilly and coastal areas of Côte d’Ivoire.

The integrated use of regional environmental resources presents the potential for sustainable development of Côte-d’Ivoire. This requires regular monitoring of the environmental conditions of the country using satellite images. In data science, processing satellite images using advanced programming methods is a challenge which consists of adaptation and optimisation of the programming algorithms towards spatial data modelling and analysis. In this study, we presented a solution to this problem with a straightforward yet effective approach which combines libraries of Python and R for processing RS data in a scripting framework. The use of both programming languages allowed us to use the best packages for topographic and environmental modelling separately when handling the satellite images. It also demonstrated the integration of the spatial libraries of the two scripting languages for terrain mapping and image processing aimed at environmental data analysis.

The techniques of R enabled image processing using information derived from pixels in the image scene. Spatial data were differentiated by the RGB triplet colours through plotting the colour composite of the Landsat 8 OLI/TIRS image using the R library ’raster’, which adapts the general syntax of R for spatial data processing. The colours corresponded to the composites of the RGB signals scattered from the natural and built-up environment, which resulted in various DN values for the pixel cells in a raster matrix of the image. The combinations of the three channels were tested by the R library ’terra’ as sensor bands, which displayed the image and visualised the appearance of various terrain objects using the distinct colours. We compared the correlation between the selected bands of Landsat-8 OLI/TIRS C1 in a multispectral image for Bands 1 and 2 (blue and green) and 4 and 5 (red and NIR). We then computed four selected vegetation indices—NDVI, SAVI, EVI2 and ARVI2—using the principles of map algebra based on the DNs of the reflectance values of the pixels in the selected spectral bands (NIR and red). The major effect of spectral reflectance from healthy green vegetation was visible in the NIR region of the image spectrum, with a strong NIR response rising sharply for healthy vegetation. Conversely, the areas with poor or damaged vegetation were represented by a low NIR response, which notably decreased as shown by the vegetation indices and with corresponding lower values for the NDVI.

The algorithms of the EarthPy library of Python were used for terrain modelling based on the high-resolution SRTM DEM. The upscaled mapping was shown at the two hierarchical levels: regional environmental analysis of the Yamoussoukro surroundings and local topographic modelling of the Kossou Lake. Processing the topographic data for visualisation of the slope, aspect, hillshade and elevation was based on interpolation of the cell values in a raster matrix grid to spatial polygons that corresponded to the diverse terrain structures. We showed these methods in a series of scripts with detailed explanations of their functionality and the effects of changed parameters on the image. The main advantage of Python for spatial modelling consists of the machine-based automation of data processing. Specifically, it derives information from the satellite images in an efficient way using a straightforward syntax, which enables handling complex RS data. The nonlinear environmental and topographic links among the near-surface processes could be depicted from raster matrices of the satellite images using programming algorithms. This enabled us to highlight the complex relationships between the objects forming the terrain shape and the diverse landscapes on Earth.

A wide variety of Earth’s features can be mapped using satellite images: climate change, urban sprawl, degradation of vegetation, deforestation, topographic variations or land cover change. At the same time, the analysis of the complexity of a particular phenomenon requires advanced methods of data processing to reveal the links and correlations among the objects and to map spatial patterns. For instance, topographic variations may be caused by geologic effects and anthropogenic activities, or the environmental dynamics of the Earth’s landscapes may involve nonlinear long-term and short-term variations. The comprehensive analysis of such processes requires advanced automatic processing of large datasets, where a conventional GIS is not effective enough. At the same time, due to the high level of spatial detail, high-resolution satellite images require efficient algorithms for data handling, including image resampling, band composition, statistical processing and map algebra for calculation of the vegetation indices. This requires the advanced methods for image processing made possible by programming languages.

As we demonstrated, the effective way to handle this issue includes the use of high-level programming languages such as Python and R. Scripting libraries are effective for automatic image processing and retrieval of spatial information from satellite images. The advanced algorithms of Python and R can be successfully used for processing Earth observation data and geospatial modelling in a robust, automatic and effective way. The application of programming approaches for terrain analysis and environmental monitoring is successful in dealing with multispectral RS data. The advantages of Python and R in image processing include scripting algorithms which ensure automation of RS data processing. Compared with GISs, the functionality of Python and R were proven to have robust handling of the geospatial data due to the high speed and effectiveness of code and repeatability of the scripts, which outperformed the existing state-of-the-art approaches of the traditional software.

In future works, we plan to use combinations of other libraries of Python and R for higher-resolution images, such as Sentinel-2A or SAR data for environmental mapping of the tropical forest regions of Africa. Specifically, we intend to focus on the long-term analysis of the environmental dynamics using time series of satellite images. Spatial modelling of the large series of RS data involves challenging problems with regard of their processing. For example, it is computationally challenging to deal with big data and requires complex methods of data handling. To this end, we plan to develop effective approaches to process large spatiotemporal datasets using scripting languages. Our next work will continue the development of the advanced cartographic methods for processing satellite images with programming languages.

## Figures and Tables

**Figure 1 jimaging-08-00317-f001:**
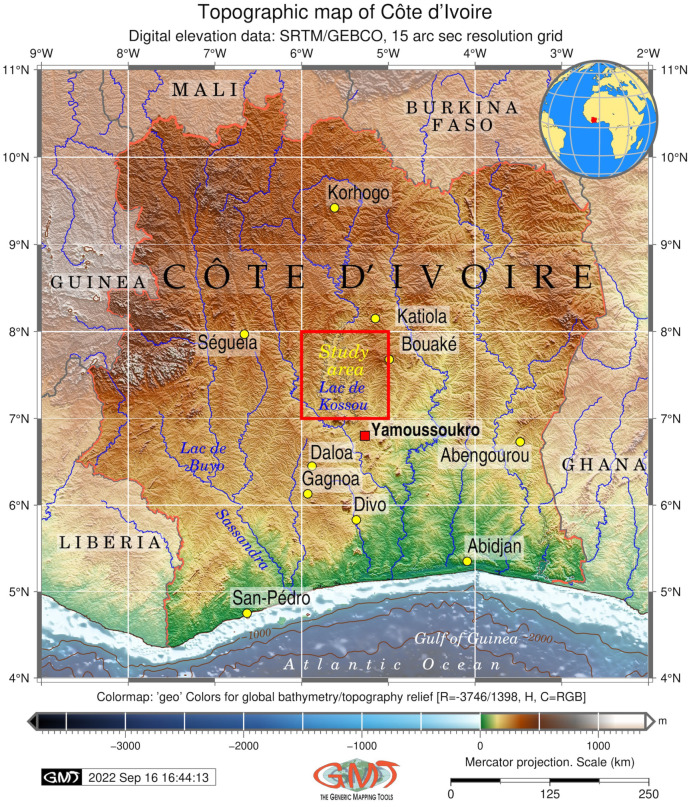
Topographic map of Côte d’Ivoire. Mapping software: Generic Mapping Tools (GMT) scripting toolset. Data source: GEBCO/SRTM. Cartography: authors.

**Figure 2 jimaging-08-00317-f002:**
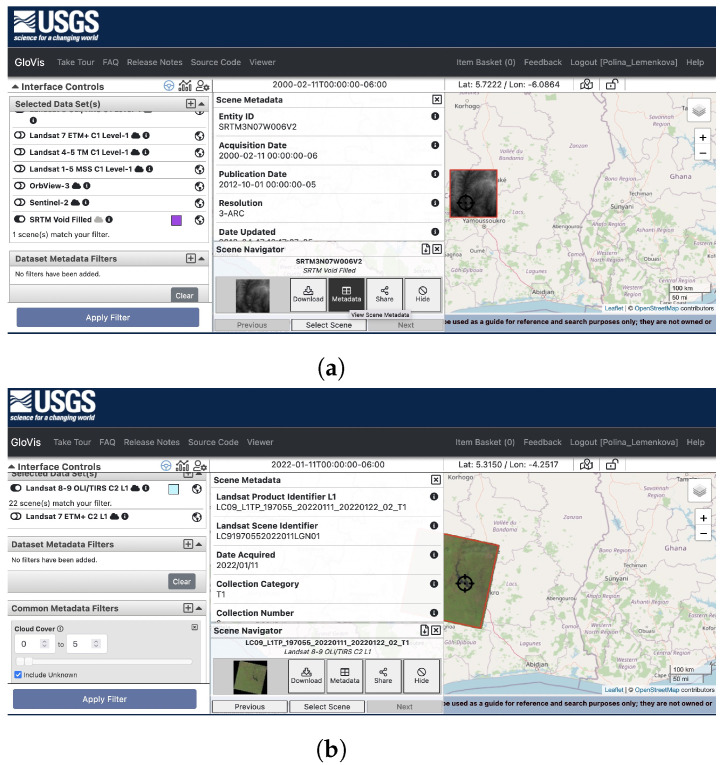
Data captured from the USGS Global Visualization Viewer (GloVis)’s satellite image repository, accessing remote sensing data from Landsat 9 OLI/TIRS C2 L1 and SRTM DEM for terrain analysis. (**a**) Collected from Landsat 9 OLI/TIRS C2 L1. (**b**) Collected from SRTM void-filled DEM.

**Figure 3 jimaging-08-00317-f003:**
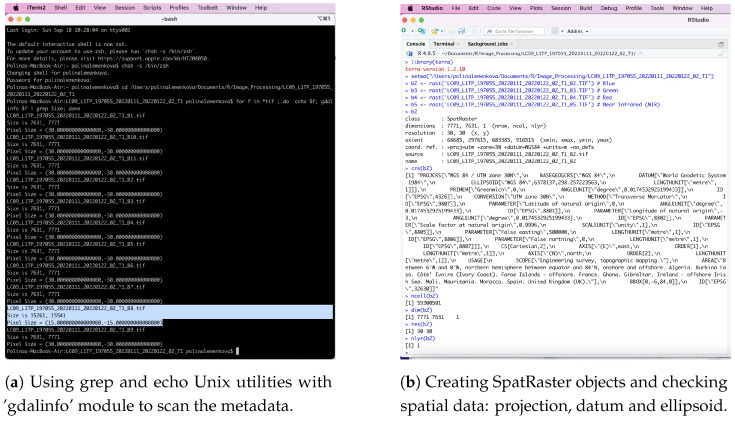
Inspecting the geometry of Landsat bands with (**a**) 30-pixel resolution for all the bands, except for Band 8 (panchromatic) with a 15-pixel resolution, checking image properties in R package ’terra’ and (**b**) for single Landsat-8 OLI/TIRS C1 bands (1:11).

**Figure 4 jimaging-08-00317-f004:**
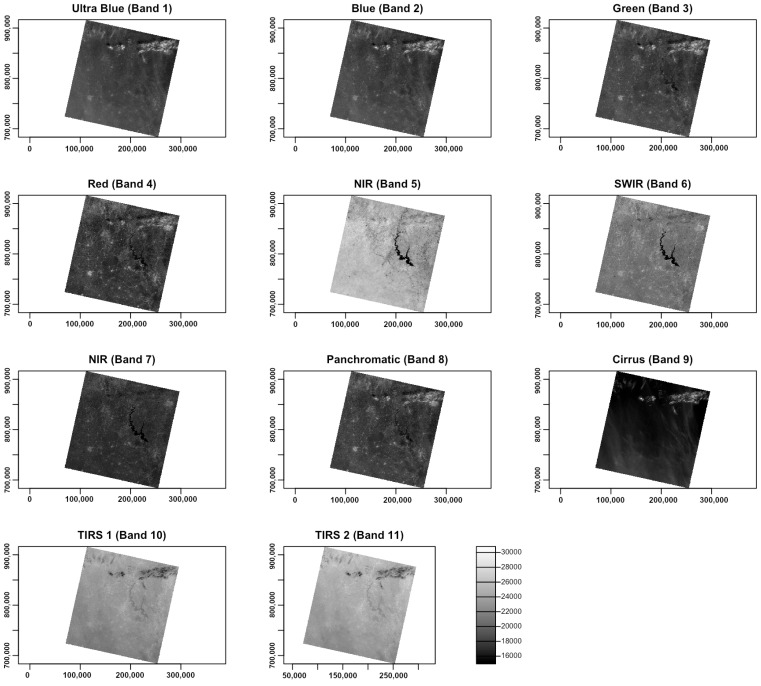
Original individual 11 layers (raw bands) of the multi-spectral image Landsat-9 OLI/TIRS C1 ‘LC09_L1TP_197055_20220111_20220122_02_T1’ in greyscale mode: ultra blue (Band 1), blue (Band 2), green (Band 3), red (Band 4), near infrared (NIR) (Band 5), Shortwave Infrared (SWIR) 1 (Band 6), Shortwave Infrared (SWIR) 2 (Band 7), panchromatic (Band 8), cirrus (Band 9), Thermal Infrared (TIRS) 1 (Band 10) and Thermal Infrared (TIRS) 2 (Band 11), plotted with R.

**Figure 5 jimaging-08-00317-f005:**
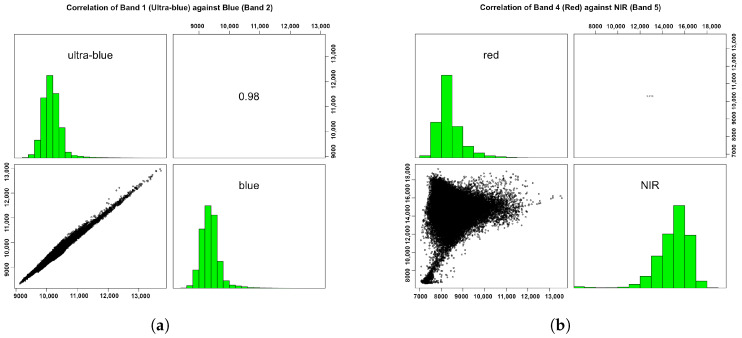
Correlation between the selected Landsat-8 OLI/TIRS C1 bands in a multispectral image *LC09_L1TP_197055_20220111_20220122_02_T1_B*. (**a**) Band 1 against Band 2. (**b**) Band 4 against Band 5.

**Figure 6 jimaging-08-00317-f006:**
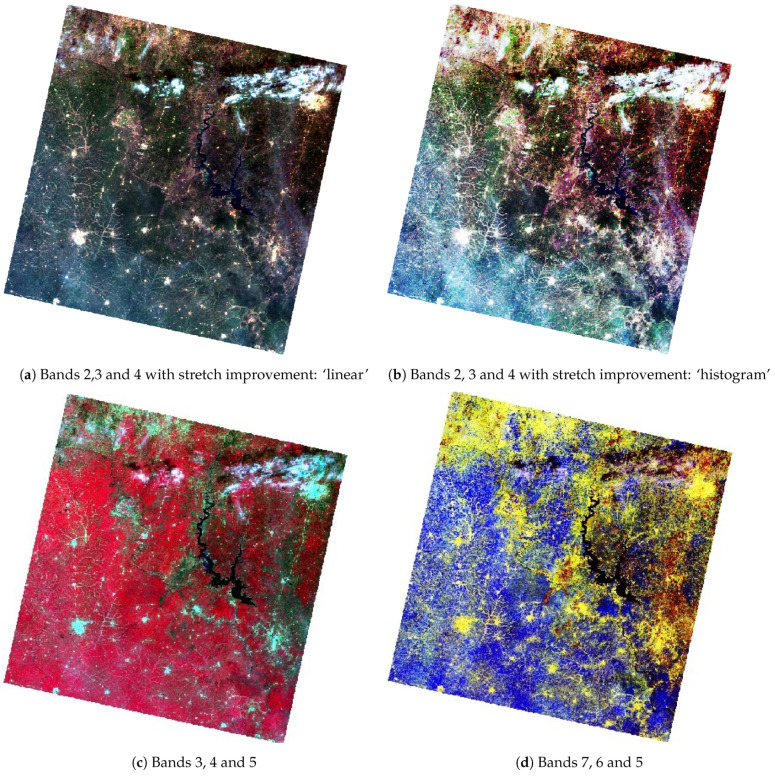
Natural color composites (**a**,**b**). False color composites (**c**,**d**). Landsat 9 OLI image of Kossou Lake region and Yamoussoukro, Côte d’Ivoire. Mapping: RStudio. Source: authors.

**Figure 7 jimaging-08-00317-f007:**
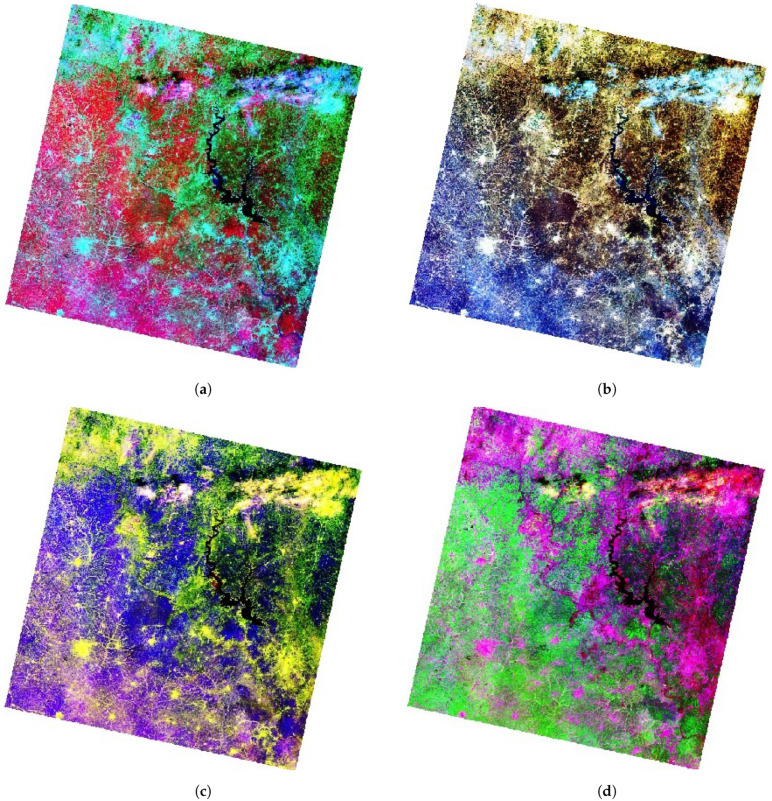
False color composite visualisation of Landsat 8 OLI image of Kossou Lake surroundings in Côte d’Ivoire. Mapping: RStudio. Source: authors. (**a**) Bands 5, 7 and 3. (**b**) Bands 7, 4 and 2. (**c**) Bands 5, 4 and 3 (color infrared). (**d**) Bands 6, 5 and 4.

**Figure 8 jimaging-08-00317-f008:**
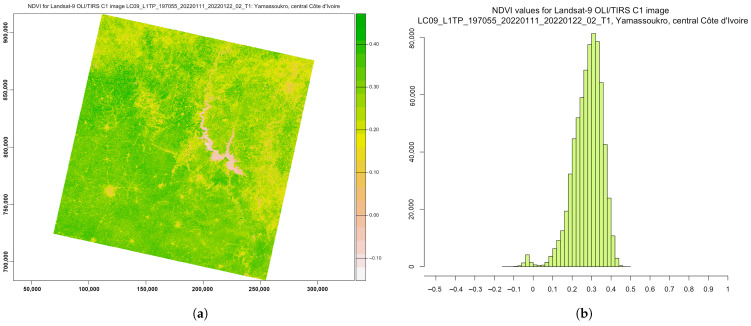
NDVI computed for multispectral Landsat-9 OLI/TIRS image. (**a**) NDVI ratio-based vegetation index. (**b**) Data distribution within the computed NDVI.

**Figure 9 jimaging-08-00317-f009:**
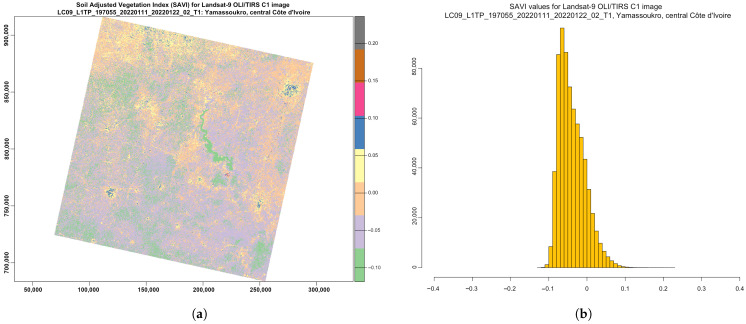
SAVI computed for multispectral image *LC09_L1TP_197055_20220111_20220122_02_T1_B*. (**a**) SAVI ratio-based vegetation index. (**b**) Data distribution within the computed SAVI.

**Figure 10 jimaging-08-00317-f010:**
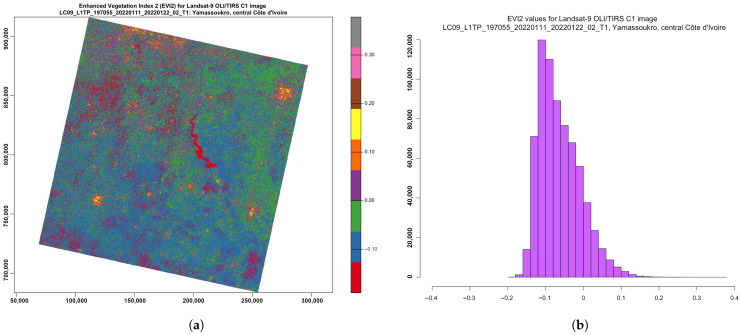
EVI2, computed for multispectral image *LC09_L1TP_197055_20220111_20220122_02_T1_B*. (**a**) EVI2 ratio-based vegetation index. (**b**) Data distribution within the computed EVI2.

**Figure 11 jimaging-08-00317-f011:**
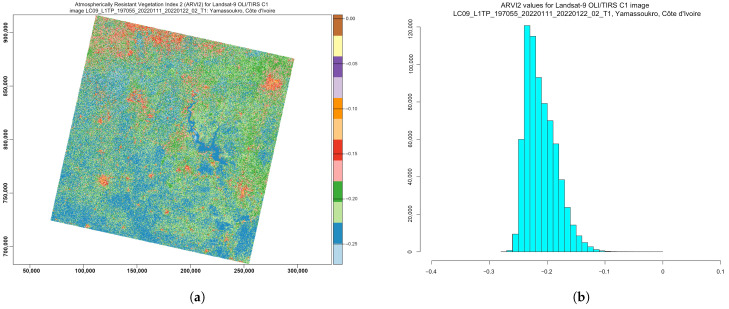
Atmospherically Resistant Vegetation Index 2 (ARVI2) computed for multispectral image *LC09_L1TP_197055_20220111_20220122_02_T1_B*. (**a**) ARVI2 ratio-based vegetation index. (**b**) Data distribution within the computed EVI2.

**Figure 12 jimaging-08-00317-f012:**
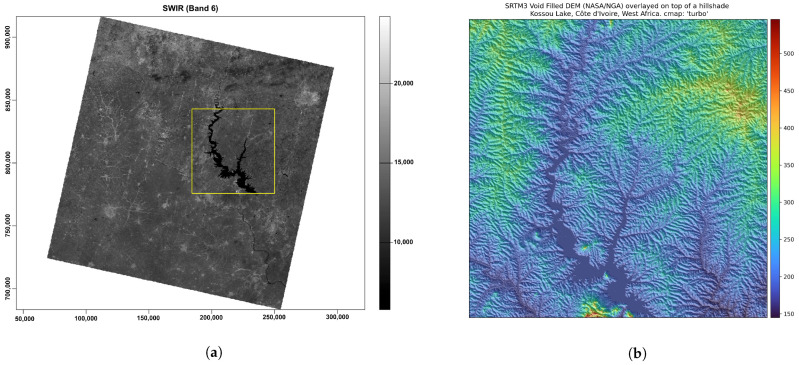
(**a**): Landsat OLI/TIRS image with enlarged fragment of Kossou Lake, using grayscale visualisation of Band 6 (SWIR). Mapping: R. (**b**) Enlarged fragment of the cropped SRTM3 void-filled DEM overlayed on top of the hillshade. Cmaps: ‘turbo’ and ‘gist_yarg’. Mapping: Python.

**Figure 13 jimaging-08-00317-f013:**
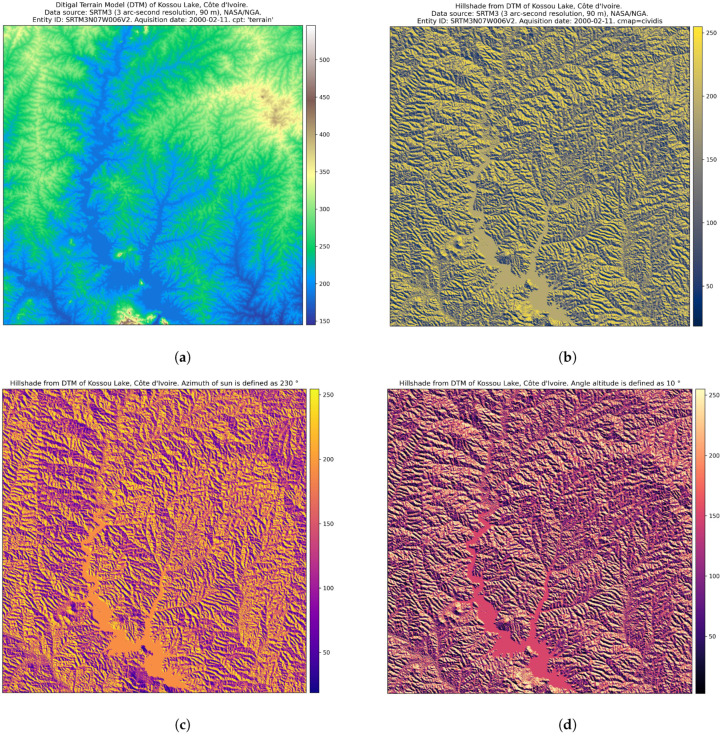
Terrain modelling for Kossou Lake area, Côte d’Ivoire. Mapping: Python. Source: authors. (**a**) DTM of Kossou Lake, Côte d’Ivoire; (**b**) Hillshade from DTM of Kossou Lake, Côte d’Ivoire. (**c**) Azimuth of Sun: 230°. Hillshade from DTM. (**d**) Sun angle altitude: 10°. Hillshade from DTM.

## Data Availability

The open GitHub repository with Python and R scripts used for image processing in this study can be found at https://github.com/paulinelemenkova/Cote_d_Ivoire_R_Python_Listings (accessed on 2 November 2022).

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
