# Peer review of "Satellite Image Processing by Python and R Using Landsat 9 OLI/TIRS and SRTM DEM Data on Côte d’Ivoire, West Africa"

_2313-433X, 2022, doi:10.3390/jimaging8120317_

Round 1

Reviewer 1 Report

The work is interesting, has a clear degree of originality, and is appropriate for publication in the journal after performing a major and very careful revision. Nevertheless, it needs some further improvements. In general, there are still some occasional grammar errors throughout the manuscript, especially the article "the," "a," and "an" are missing in many places; please make spellchecking in addition to these minor issues. The reviewer has listed some specific comments that might help the authors further enhance the manuscript's quality.

  1. Specific Comments

·        Overall, the Abstract section is not giving any information about methodology, results, conclusion, and recommendations as it should be with clear. I suggest the authors to remove generic lines and present the strong statements and novelty of article. The abstract written by qualitative sentences. It is need to modify and rewrite based on the most important quantity results from this research. The abstract should be redesigned. You should avoid using acronyms in the abstract and insert the work's main conclusion.

·        You have used many abbreviations in the text. From this perspective, an Index of Notations and Abbreviations would be beneficial for a better understanding of the proposed work. Furthermore, please check carefully if all the abbreviations and notations considered in work are explained for the first time when they are used, even if these are considered trivial by the authors. The paper should be accessible to a wide audience. Furthermore, it will make sense to include also the notations in this index.

  • The objectives should be more explicitly stated.
  • The Introduction section must be written on more quality way. The research gap should be delivered on more clear way with directed necessity for the conducted research work.
  • Please elaborate on the introduction section. The following literature may be helpful in this regard: << Detection of Water Spread Area Changes in Eutrophic Lake Using Landsat Data>>, << DEM resolution effects on machine learning performance for flood probability mapping>> you may consider additional references as well.
  • What is the novelty of this work?
  • It is better to improve your contributions which are not so clear to show the advantage of your work.

·        The novelty of this work must be clearly addressed and discussed in Introduction section.

  • The methodology limitation should be mentioned.

Many equations are presented in the paper, and most look OK. However, please check carefully whether all equations are necessary and whether the quantities involved are properly explained. Also, some equations need references.

  • Results
  • This section is well written.

  • Discussion
  • Overall, the discussion part is weak. The Discussion should summarize the manuscript's main finding(s) in the context of the broader scientific literature and address any study limitations or results that conflict with other published work.

  • Conclusion
  • Some future works should be added to your conclusion.

Author Response

Dear Editors of the Journal of Imaging,

Please find attached the revised version of the paper. We have carefully followed the comments and critical suggestions of the reviewers and corrected the manuscript accordingly.

All the changes in the text are colored yellow, for the convenience of track changes. Changes in the Abstract are coloured blue.

The replies to the comments in the reviewer’s reports are listed below.

Using the opportunity, we thank the reviewers for their careful reading and critical remarks which improved the initial version of the manuscript.

With kind regards, - Authors (Polina Lemenkova and Olivier Debeir).

02.11.2022.

Reviewer 1

No

Reviewer’s Comments

Author’s actions

1

Does the introduction provide sufficient background and include all relevant references? – Can be improved

The Introduction has been updated and partially reformulated; some paragraphs have been moved; new sentences added; rewording of some phrases has been done where necessary.

2

Are all the cited references relevant to the research? – Can be improved

The suggested two articles are added and cited, as well as some more items. The References section is updated with seven more new entries (see also comment #11).

3

Are the methods adequately described? – Can be improved

The Methodology section is improved: updated the text overall, improved small minor misprints, added some more comments to the codes, added a paragraph on limitations, as suggested in comment #15.

4

Are the results clearly presented? – Can be improved

The Results and Discussion section is updated and improved. Many new sentences are included and/or partially rewritten; some small minor improvements in the text are made where required.

5

Are the conclusions supported by the results? – Can be improved

The Conclusions are rewritten and updated. Many sentences are included; last paragraph regarding future works and perspectives of scripting languages for satellite image processing is included.

6

The work is interesting, has a clear degree of originality, and is appropriate for publication in the journal after performing a major and very careful revision. Nevertheless, it needs some further improvements. In general, there are still some occasional grammar errors throughout the manuscript, especially the article "the," "a," and "an" are missing in many places; please make spellchecking in addition to these minor issues. The reviewer has listed some specific comments that might help the authors further enhance the manuscript's quality.

The manuscript is proofread throughout. We have corrected all occasional typesetting misprints and minor grammar mistakes (spelling, punctuation) where necessary. Grammar errors are corrected and misprints are checked everywhere in the text.

7

Overall, the Abstract section is not giving any information about methodology, results, conclusion, and recommendations as it should be with clear. I suggest the authors to remove generic lines and present the strong statements and novelty of article. The abstract written by qualitative sentences. It is need to modify and rewrite based on the most important quantity results from this research. The abstract should be redesigned. You should avoid using acronyms in the abstract and insert the work's main conclusion.

The Abstract is modified and rewritten as suggested, using the most important research results. General statements are removed, instead, more specifics regarding the vegetation indices in Côte d'Ivoire is added. The comparison of indices is briefly provided as well as the comments of Python and R functionality. The Abstract is largely rewritten and modified significantly. The acronyms are deleted from the Abstract; the remaining are explained explicitly (e.g., Digital Elevation Model (DEM)); the main conclusions are inserted.

8

You have used many abbreviations in the text. From this perspective, an Index of Notations and Abbreviations would be beneficial for a better understanding of the proposed work. Furthermore, please check carefully if all the abbreviations and notations considered in work are explained for the first time when they are used, even if these are considered trivial by the authors. The paper should be accessible to a wide audience. Furthermore, it will make sense to include also the notations in this index.

Abbreviations are inserted in the manuscript on p. 26 with acronyms and full names for each entry. We checked the appearance of each abbreviated words for its 1st mention in the text and corrected where necessary. Several missing entries are added (Landsat TM, DN, SAGA, etc). The notations are explained in the codes and in the comments to equations (e.g., y is the quotient derived from the components of atmospheric reflectance in Eq. 5); some of the notations in Python code are self-explaining, e.g., ‘hist’ stands for a histograms, ‘plot’ is plotting a figure, etc. Some more small explanations are added in the text.

9

The objectives should be more explicitly stated.

Added in the paragraph: “The objectives of our study were to use the advanced functionality of several libraries of R and Python for treatment of image bands as variables in single scripts for processing each task separately: terrain modelling, computing vegetation indices through map algebra, combination of Landsat bands. Our method depends on using flexible algorithms <...> in contrast to the previous work, we used advanced libraries of Python and packages of R languages for processing high-resolution satellite images aimed at environmental assessment and mapping of C\^ote d'Ivoire and specifically, modelling the terrain in the surroundings of Kossou Lake, to the north of Yamoussoukro.

10

The Introduction section must be written on more quality way. The research gap should be delivered on more clear way with directed necessity for the conducted research work.

The Introduction section is revised, with many new sentences added, others rewritten and partially modified. The proofreading is made overall. The research gap is addressed in the phrase “Moreover, the GIS methods essentially need time and manual operational routine to arrive completion” and mentioned disadvantages of the state-of-the-art methods of GIS. Then we described the advantages of the presented two approaches – Python and R open source programming tools: “In this paper, two approaches are proposed to solve the problem of satellite image processing by using and R programming languages. As the first approach <...> The combination of several libraries can efficiently utilize the properties and functionality of each to achieve effective image processing for geographical analysis.

11

Please elaborate on the introduction section. The following literature may be helpful in this regard: << Detection of Water Spread Area Changes in Eutrophic Lake Using Landsat Data>>, << DEM resolution effects on machine learning performance for flood probability mapping>> you may consider additional references as well.

The list of references is updated with included suggested items (DOI: 10.3390/s22186827 cited in phrase “Examples of other environmental indices include, for instance Normalized Difference Water Index (NDWI), modified normalized difference water index (MNDWI), and water ratio index (WRI)” and DOI: 10.1016/j.jher.2021.10.002 (Added in phrase "The DEM with high spatial resolution is one of the most crucial base layer factors for environmental modelling..") and some more:

1) DOI: 10.1007/s12665-012-1622-2 (Doumouya et al. 2012)

2) DOI: 10.1007/s10531-007-9292-1 (Hennenberg et al. 2008)

3) DOI: 10.1007/s10668-020-00639-8

(El-Shahat et al. 2021)

4) DOI: 10.1007/s11273-015-9474-7

(Tang et al. 2016)

5) DOI: 10.1007/s10661-008-0649-z

(Affian et al. 2009)

Literature is cited in the Introduction section accordingly. The phrases containing new citations are highlighted.

12

What is the novelty of this work?

Added a special passage in the end of Introduction: “To the best of our knowledge, no attempts have been made to utilize high-level scripting languages, such as Python and R, for mapping Côte d'Ivoire using satellite image processing. A related works to ours are some previous works [85–89] in which the traditional GIS methods are applied to process spatiotemporal information for environmental mapping <...> interface of the existing programs with restricted functionality of the state-of-the-art cartographic software.” and “The advanced functionality of several packages of R and libraries of Python was used <...> Kossou Lake, to the north of Yamoussoukro

13

It is better to improve your contributions which are not so clear to show the advantage of your work.

The notes on contributions are added in the last paragraph of Introduction: “Our main contributions can be summarized as follows: (1) the environmental mapping of Côte d’Ivoire based on modelling and visualization of several vegetation indices using 'terra' package; (2) the R-based approach for processing satellite images <...>; (3) using EarthPy-based algorithms for deriving topographic information for spatial analysis of the heterogeneous landscape of Kossou Lake region to <...> automatically; (4) mapping spatial data by scripting libraries that improve on those obtained by state-of-the-art GIS <...>

14

The novelty of this work must be clearly addressed and discussed in Introduction section.

Yes, added a special paragraph; answered in comment 12 above. We stressed the novel approach of cartographic data processing by integrated using of 2 programming languages specifically for the region of Côte d'Ivoire which his much rarely studied compared to other West African countries, e.g., such as Ghana, or Senegal. We used advanced libraries of Python and packages of R for processing high-resolution satellite images aimed at environmental assessment and mapping.

15

The methodology limitation should be mentioned.

Added a passage regarding the limitation, starting with "Our current limitations is that we assume the data to be of high resolution <...> Satellite pour l’Observation de la Terre (SPOT 7) with up 721 to 1.5 m spatial resolution in panchromatic and multispectral channels."

16

Many equations are presented in the paper, and most look OK. However, please check carefully whether all equations are necessary and whether the quantities involved are properly explained.

Yes, the equations are necessary, because they clearly indicate the formulae for band combinations in each case: Eq. 1 explains the ratio of bands for NDVI, Eq. 2 – for SAVI, Eq. 3 and 4 – for EVI and EVI2, Eq. 5 and 6 – for ARVI and ARVI2, respectively, All these indices were then visualized and plotted, so the information on combinations of channels (bands) used for plotting is crucial. Some comments are added, as well as citations to the original source of formulae.

17

Also, some equations need references.

References are added for all the equations:

Eq. 1 (NDVI) – [130]; Eq. 2 (SAVI)– [131]; Eq. 3 and 4 (for EVI and EVI2) – [132]; Eq. 5 and 6 (ARVI and ARVI2) – [133].

18

Results. This section is well written.

Ok. This section is checked for proofreading; some minor textual corrections are made.

19

Discussion. Overall, the discussion part is weak. The Discussion should summarize the manuscript's main finding(s) in the context of the broader scientific literature and address any study limitations or results that conflict with other published work.

The Discussion is elaborated in the results and Discussion section, where more information is added on the specifics of the vegetation indices, i.e., why SAVI, ARVI or EVI indices are better compared to the NDVI and how they can be applicable specifically for tropical African region of Côte d'Ivoire. For instance, The effects from soil and atmospheric factors that were excluded in these indices, compared to NDVI, are pointed, so that SAVI is more adjusted to the agricultural monitoring. At the same time, because Côte d'Ivoire is a highly productive agricultural region (more information is in Study area section, e.g. cocoa producer), the agricultural monitoring is of importance for this country.

20

Conclusion. Some future works should be added to your conclusion.

Added in the last paragraph: “In future work, we plan to utilize combinations of other libraries of Python and R for spatial datasets, such as Sentinel-2A, SAR data for retrieving environmental parameters aimed at tropical forest mapping of Africa. Specifically, we intend to focus <...>. To this end, we plan to develop effective approaches to map and model complex and large spatio-temporal datasets using scripting languages, which creates a valuable approach in the context of big Earth observation data and contributes to the development of contemporary cartography and mapping”.

Reviewer 2 Report

The authors proposed a method for processing satellite images to create maps using Python and R. In particular, the authors addressed the problem of deriving information on vegetation coverage and relief parameters. In the presented case study they relied on Landsat 9 OLI/TIRS and SRTM DEM datasets covering the territory of Côte d’Ivoire, West Africa. The developed scripting algorithms of Python and R demonstrated high quality and performance, in contrast to the existing GIS approaches for remote sensing data processing. 

The manuscript is well-written and the results are promising, so my advice is to be accepted for publishing in the journal. Since the manuscript includes several listings with the contributed scripting algorithms as well as obtained visualization results, my advice for authors is to create a GitHub repository with appropriate files and share the link in the paper.

Author Response

Dear Editors of the Journal of Imaging,

Please find attached the revised version of the paper. We have carefully followed the comments and critical suggestions of the reviewers and corrected the manuscript accordingly.

All the changes in the text are colored yellow, for the convenience of track changes. Changes in the Abstract are coloured blue.

The replies to the comments in the reviewer’s reports are listed below.

Using the opportunity, we thank the reviewers for their careful reading and critical remarks which improved the initial version of the manuscript.

With kind regards, - Authors (Polina Lemenkova and Olivier Debeir).

02.11.2022.

Reviewer 2

No

Reviewer’s Comments

Author’s actions

1

The authors proposed a method for processing satellite images to create maps using Python and R. In particular, the authors addressed the problem of deriving information on vegetation coverage and relief parameters. In the presented case study they relied on Landsat 9 OLI/TIRS and SRTM DEM datasets covering the territory of Côte d’Ivoire, West Africa. The developed scripting algorithms of Python and R demonstrated high quality and performance, in contrast to the existing GIS approaches for remote sensing data processing.

General comment (no actions are required).

2

The manuscript is well-written and the results are promising, so my advice is to be accepted for publishing in the journal. Since the manuscript includes several listings with the contributed scripting algorithms as well as obtained visualization results, my advice for authors is to create a GitHub repository with appropriate files and share the link in the paper.

Many thanks for endorse of our manuscript and support!

We created the open GitHub repository with all the 10 codes of Python and R used in this study for image processing (Listings 1 to 10 in the paper): https://github.com/paulinelemenkova/Cote_d_Ivoire_R_Python_Listings

The link to this folder is provided both in the text of the manuscript and in the Data Availability statement.
